# Personalized and muscle-specific OXPHOS measurement with integrated CrCEST MRI and proton MR spectroscopy

Ryan R. Armbruster [1,6], Dushyant Kumar[1,6], Blake Benyard[1], Paul Jacobs [1], Aditi Khandavilli[2], Fang Liu[3], Ravi Prakash Reddy Nanga[1], Shana McCormack[4], Anne R. Cappola[5], Neil Wilson[1] & Ravinder Reddy [1] ✉

Creatine chemical exchange saturation transfer (CrCEST) MRI is an emerging high resolution and noninvasive method for measuring muscle specific oxidative phosphorylation (OXPHOS). However, CrCEST measurements are sensitive to changes in muscle pH, which might confound the measurement and interpretation of creatine recovery time ($\tau_{Cr}$). Even with the same prescribed exercise stimulus, the extent of acidification and hence its impact on $\tau_{Cr}$ is expected to vary between individuals. To address this issue, a method to measure pH pre- and post-exercise and its impact on CrCEST MRI with high temporal resolution is needed. In this work, we integrate carnosine [1]H- magnetic resonance spectroscopy (MRS) and 3D CrCEST to establish "mild" and "moderate/intense" exercise stimuli. We then test the dependence of CrCEST recovery time on pH using different exercise stimuli. This comprehensive metabolic imaging protocol will enable personalized, muscle specific OXPHOS measurements in both healthy aging and myriad other disease states impacting muscle mitochondria.

In energy metabolism, phosphocreatine (PCr) plays an essential role as a phosphate shuttle for depleted adenosine triphosphate (ATP). PCr has a high concentration in muscle (~20–35 mM), and in healthy participants it is quickly catalyzed through a reversible enzyme, creatine kinase (CK). During exercise, the forward reaction occurs, where creatine kinase transfers a phosphate group to ADP which forms free creatine and ATP[1].

$$ADP + PCr + H + \underset{}{\overset{CK}{\rightleftharpoons}} Cr + ATP \tag{1}$$

The reverse reaction occurs during rest, and its kinetics can be quantified using the phosphocreatine ($1/\tau_{PCr}$) or creatine ($1/\tau_{Cr}$)

recovery rate. This measurement is strongly correlated with oxidative phosphorylation (OXPHOS) capacity or ATP synthesis, whereby it is known that longer recovery times of $\tau_{PCr}$ or $\tau_{Cr}$ are characteristics of decreased OXPHOS capacity[2].

Variations in creatine kinase (CK) are present in various pathologies such as mitochondrial disorders[2], cardiovascular disease[3], and Becker's and Dushenne's muscular dystrophies[4-6]. Frequently, muscle biopsies are performed to test for disease, which are not only invasive but also difficult to perform in participants longitudinally, e.g., in the course of natural history studies or clinical trials.

To mitigate the need for such invasive testing, magnetic resonance spectroscopy (MRS) techniques such as phosphorous MRS ([31]P-MRS) have traditionally been employed to measure high-energy

[1]Department of Radiology, Center for Advanced Metabolic Imaging in Precision Medicine, University of Pennsylvania, Philadelphia, PA 19104, USA. [2]Department of Biology, Department of Nutrition and Science, Cornell University, Ithaca, NY 14850, USA. [3]Department of Biostatistics, Epidemiology, and Informatics, Perelman School of Medicine, University of Pennsylvania, Philadelphia, PA 19104, USA. [4]Neuroendocrine Center, Division of Endocrinology and Diabetes, Children's Hospital of Philadelphia, Philadelphia, PA 19104, USA. [5]Division of Endocrinology, Diabetes, and Metabolism, University of Pennsylvania, Philadelphia, PA 19104, USA. [6]These authors contributed equally: Ryan R. Armbruster, Dushyant Kumar. ✉e-mail: krr@pennmedicine.upenn.edu

phosphate compounds, such as PCr, inorganic phosphate (Pi) and ATP, as well as to track intracellular pH[7–10]. However, the clinical utility of [31]P-MRS is limited due to its low sensitivity, limited spatial resolution, and requirements for multinuclear coils. Proton MRS ([1]H-MRS) is an alternative approach that has been used in attempt to measure PCr and Cr. However, both have aliphatic protons with similar chemical shifts which make quantifying the peaks separately difficult. PCr and Cr measurement using [1]H-MRS has only been performed in animals at ultra-high field strengths (9.4 T)[11]. It has been shown that creatine's guanidinium protons exchange with bulk water at a rate of $950 \pm 100\,\mathrm{s}^{-1}$ [12]. Recently, a proton based MRI measurement of creatine was developed that exploits the chemical exchange saturation transfer (CEST) effect from guanidinium protons of creatine (CrCEST) with water protons[13,14]. In this method, a long, low power, frequency selective radiofrequency is used to saturate guanidinium protons of creatine (-1.8 ppm downfield from water). The saturated magnetization from guanidinium protons exchange with bulk water, and, during the course of saturation, the bulk water magnetization decreases in proportion to creatine concentration. This decreased magnetization is readout by a selected imaging sequence. Indeed, 2D/3D CrCEST has been successfully used to measure muscle-specific creatine recovery time ($\tau_{Cr}$) -values with adequate spatial resolution, volume coverage, and temporal resolution (-30 s) for studies in humans[15]. The sensitivity improvement was shown to be three orders of magnitude higher than [31]P-MRS[14].

Despite these advantages, a drop in intracellular pH leads to significant biases in estimated $\tau_{Cr}$-values[13]. Specifically, acidosis results in a reduction of CrCEST asymmetry as well as a reduction in the forward reaction of CK enzyme kinetics[16]. However, the relationship between $\tau_{Cr}$ and pH in physiological conditions is still not well understood. To date, most studies using CrCEST in skeletal muscle have employed exercise stimuli designed to be mild and easily accomplished by participants. However, individuals vary with respect to their exercise capacity and performance, thus identical exercise stimuli might produce differences in post-exercise pH, although post-exercise pH was not typically measured in prior CrCEST studies because it difficult to measure muscle-specific pH in vivo with [31]P-MRS. Therefore, the extent of acidification and hence its impact on $\tau_{Cr}$ may not have been standardized between participants.

[1]H-MRS can be used to measure post-exercise pH in place of [31]P spectroscopy because [1]H-MRS allows for concurrent CrCEST and pH measurements within the same exercise bout and MRI scanning session. Proton spectroscopy ([1]H-MRS) detects shifts in the imidazole moieties of carnosine (beta-alanyl-L-histidine) which acts primarily as a pH buffer in musculature[17]. Its C2 and C4 protons have physiological pK$_a$'s, which are sensitive to changes in pH[18,19], and their sensitivity to pH correlates well with [31]P results and lactate measurements[20–22].

In this work, we integrated [1]H-MRS of carnosine and 3D CrCEST to establish mild and moderate/intense exercise stimuli. Specifically, we determined the exercise stimulus as a proportion of the individual-specific maximal voluntary contraction (MVC) that achieved a non-significant (≤0.1 unit) change in pH (for mild exercise) versus a significant (>0.1 units) change in pH (for moderate/intense exercise). We then tested the dependence of CrCEST recovery time on pH using the different exercise stimuli.

## Results

### Visualizing 3D CrCEST

Figure 1 shows the anatomical images and processed CrCEST maps in stacked form. As seen in the stacked CrCEST maps, all the slices exhibit CrCEST asymmetry contrast for the mild and moderate/intense exercise bouts. In the moderate/intense exercise column, regions outside the gastrocnemius are also producing strong CrCEST signal. This portion of the calf belongs to the soleus, which was not analyzed as its usage is not uniform across participants during plantar flexion exercise. All participants showed the greatest CrCEST increase in the lateral gastrocnemius (LG) and medial gastrocnemius (MG) muscle groups. Furthermore, a majority of participants had the highest CrCEST asymmetry increase in the middle slice, which corresponds to the thickest portion of their calf. Supplemental Fig. 1 shows that the cross relaxation contribution of the aliphatic protons of creatine, measured using transient nuclear Overhauser effect (tNOE), is negligible with a contrast of 0.07% from a 100 mM creatine phantom. Upon asymmetry analysis the contribution becomes 0.03%. Given that the physiological concentration of creatine is -20−35 mM, this would mean that relative cross relaxation contribution to CrCEST signal would be <0.01%. This is consistant with CrCEST work which showed no appreciable CEST signal from the creatine methyl protons at −1.7 ppm[12].

### pH shifts measured via carnosine spectroscopy

Figure 2A shows the carnosine molecule with its imidazole moiety and C2 and C4 protons marked in red. Figure 2B depicts the anatomical image of the calf with the green spectroscopic voxel shown in the MG. Across all participants, there was no consistency as to which muscle group had the largest pH change during plantar flexion. Of the 27 participants that performed mild and moderate/intense exercise, 10 exercised the MG most, 14 exercised the LG most, and 3 exercised both muscles similarly. Figure 2C tabulates the chemical shift measured in the MG for one representative participant at baseline, mild exercise, and moderate/intense exercise. The table of values (Fig. 2C) shows that the participant's pH shift of (0.03 pH units) meets our mild exercise stimulus criterion which includes pH shifts ≤0.1 pH units.

Figure 2D shows the carnosine spectrum that contains both the C2 and C4 protons, including not only the C2-H peak shifts, but also the C4-H peak due to its physiologic pK$_a$. In the mild exercise spectrum, the baseline of the spectrum is noticeably noisier due to changes in the participant's calf position post-exercise. For the same reason, the C2-H peak in the moderate/intense exercise spectrum is much broader. This did not effect the measurement of the carnosine shifts as there was sufficient SNR in each spectrum to detect the peaks. In our analysis, we only saw one participant with a moderate/intense exercise spectrum that had both a peak shift and peak splitting occur in the C2 protons. This constellation of findings is due to the muscle group exhibiting a distinct oxidative and glycolytic environment which causes carnosine

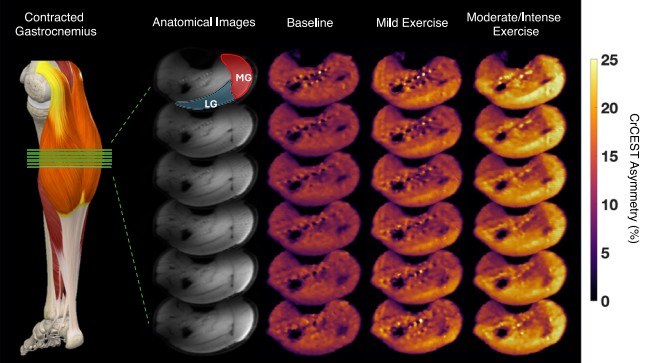

**Fig. 1 | CrCEST asymmetry maps of the gastrocnemius after different exercise bouts.** A physiological model of a contracted gastrocnemius, which is highlighted in yellow, is displayed on the far left (Complete Anatomy (3D4Medical, Elsevier)). Six green slices represent the positioning of the anatomical images. Each column displays a stacked set of slices acquired from each 3D acquisition. Slices two through six are shown, and slices one and eight were discarded due to overlap artifact. The first panel is an anatomical image. The regions of interest are segmented in red (medial gastrocnemius, MG) and blue (lateral gastrocnemius, LG). Panels two through four are the baseline, mild exercise, and moderate/intense exercise CrCEST asymmetry maps. CrCEST maps corresponding to the first time-point post-exercise are taken and displayed for the mild and moderate/intense exercise columns.

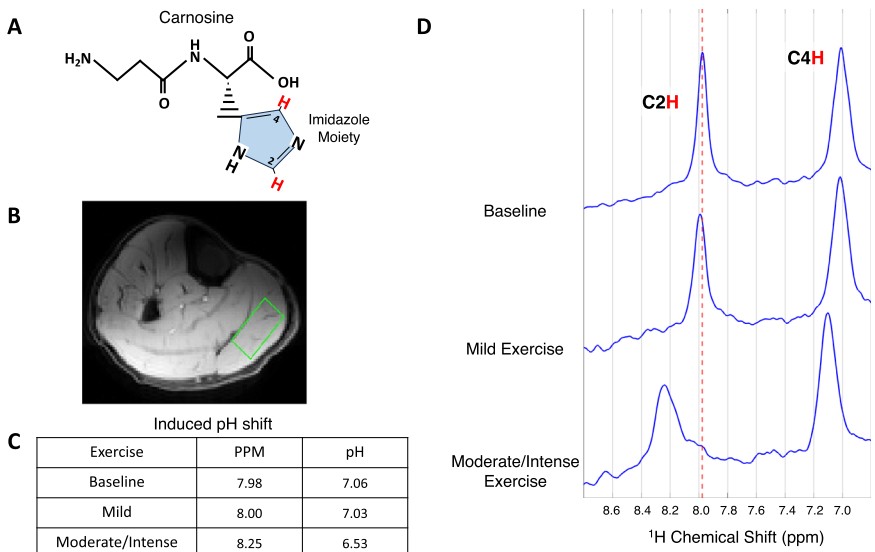

**Fig. 2 | Carnosine spectroscopy based pH shift of calf region after exercise.**
**A** The molecular structure of carnosine with its blue imidazole moiety and red C2 and C4 protons are displayed. **B** An anatomical image of the right calf with a green ROI placed in the medial gastrocnemius. **C** Resultant ppm values are displayed with the calculated pH given by the adjusted Henderson–Hasselbach equation. **D** The chemical shift at baseline and all exercise regimes is shown. The red dotted line is drawn at the peak of the C2 proton at 7.98 ppm as a reference to visualize the ppm shift relative to each exercise. Source data are provided in Source Data file.

to be affected by two different pH environments. All other participants had a resultant carnosine C2-H peak shift without peak splitting. Intense exercise peak shifts ranged from 0.06–0.72 ppm units which correspond to a pH range of 6.27–6.96. A video of a large peak shift and its subsequent recovery over 15 minutes is displayed in Supplemental Movie. 1.

**Mapping CrCEST recovery in mild and moderate/intense exercise**
Placement of the dielectric pad on the lateral gastrocnemius helped to improve relative $B_1^+$ inhomogeneity (Fig. 3). In the LG, the mean and range of the relative $B_1^+$ went from 0.68 (0.47–1.0) without pads to 1.11 (0.93–1.35). Whereas in the MG, the relative $B_1^+$ remained similar as it decreased marginally from 1.05 (0.82–1.29) to 1.00 (0.71–1.26).

Figure 4A shows a representative CrCEST time series for the same participant as in Fig. 1, after mild exercise. The left most block in Fig. 4A displays the baseline CrCEST map and regions of interest (ROIs) of the LG and MG in blue and red, respectively. Placement of the thickest portion of the calf at isocenter and padding the calf to ensure it is in isocenter in both the right-left and anterior-posterior directions helped to improve shim quality. The $\tau_{Cr}$ recovery curve in Fig. 4B shows that the lateral gastrocnemius was the most utilized muscle group post-mild exercise for this participant. During mild exercise, 5 of the 27 participants had the longest $\tau_{Cr}$ in the MG, 21 of 27 in the LG, and 1 of 27 had similar values (<5% difference) in both calf regions. The measured values for baseline and post-mild exercise CEST asymmetry values (Asym. %), as well as $\tau_{Cr}$ recovery time (s) are shown in Fig. 4C. The post-exercise CrCEST Asym. % and $\tau_{Cr}$ recovery times (s) show how divergent the LG and MG can be in terms of kinetics, despite each extensor muscle group contributing to the same exercise.

The CrCEST results of the same participant's moderate/intense exercise is detailed in Fig. 5. The data are portrayed the same way as in Fig. 4, except that in Fig. 5A, the time series of CrCEST recovery shows twelve timepoints instead of eight. A longer post-exercise time course was shown to illustrate how the moderate/intense exercise prolongs the CrCEST recovery time. In Fig. 5B, the LG and MG show a similar increase in CrCEST Asym. %. Despite having the same CrCEST Asym. %, they recover to baseline at different rates, as shown in column five of Fig. 5C. The CrCEST Asym. % recovery curve has a 45 second time gap

between the third and fourth timepoint. This extra time was necessary to obtain the post-intense exercise carnosine acquisition. Moderate/intense exercise yielded 5 of 17 exercising MG most, 9 exercising LG most, and 3 exercising both regions similarly (<5% difference).

**Healthy participants $\tau_{Cr}$ and $\Delta$pH in baseline and post-exercise**
Descriptive statistics that include the mean and standard deviation of baseline and post-exercise values for CrCEST Asym. % as well as the ppm and pH shifts are shown in Tables 1 and 2, respectively. $\tau_{Cr}$ values in Table 1 include the median and range of the datasets. The mean and standard deviation was not reported because the distributions of both mild and intense $\tau_{Cr}$ were not normally distributed. The number of participants contributing to each analysis is shown. Table 1 and 2 are then categorized by exercise stimulus and calf region. Fewer than 27 participants are shown for the mild exercise regime for four reasons: (1) two participants exhibited excessive motion during the CrCEST scans, (2) two participants had very inhomogeneous $B_1^+$ fields, (3) six participants had pH shifts >0.1 during what was supposed to be mild exercise, and (4) one participant had a minimal CrCEST asymmetry increase of <10% during mild exercise. To utilize all possible data, the six datasets described in point three were incorporated into the moderate/intense exercise participant pool. For this reason, although only 17 total datasets were planned to be included in the moderate/intense exercise analysis there are more than 17 used.

For all participants, baseline CrCEST Asym. % values were near 10% with mean $\Delta$CrCEST ranges from 3.7 - 4.7% for mild exercise and 6.7–7.0% for moderate/intense exercise, in the MG and LG, respectively. The median $\tau_{Cr}$ recovery times were near 60 s for mild exercise and nearly double for moderate/intense exercise, ~120 s. The baseline ppm and pH were physiologically normal and consistent with other measurements in the literature[20,23]. The mean mild exercise pH shift was 0.03 units for both LG and MG. The mean moderate/intense exercise pH shift was 0.30 and 0.36 in the MG and LG, respectively.

To assess the relationship of $\tau_{Cr}$ values vs. decrease in pH, a Spearman's rank correlation analysis was performed on mild exercise and moderate/intense exercise data (Fig. 6A, B). Each plot shows the Spearman's $\rho$ and $p$ value. Figure 6A displays the mild exercise participants. An x-intercept at 0.1 is drawn to visualize the cutoff for decrease in pH in the mild regime. For the mild exercise group, the

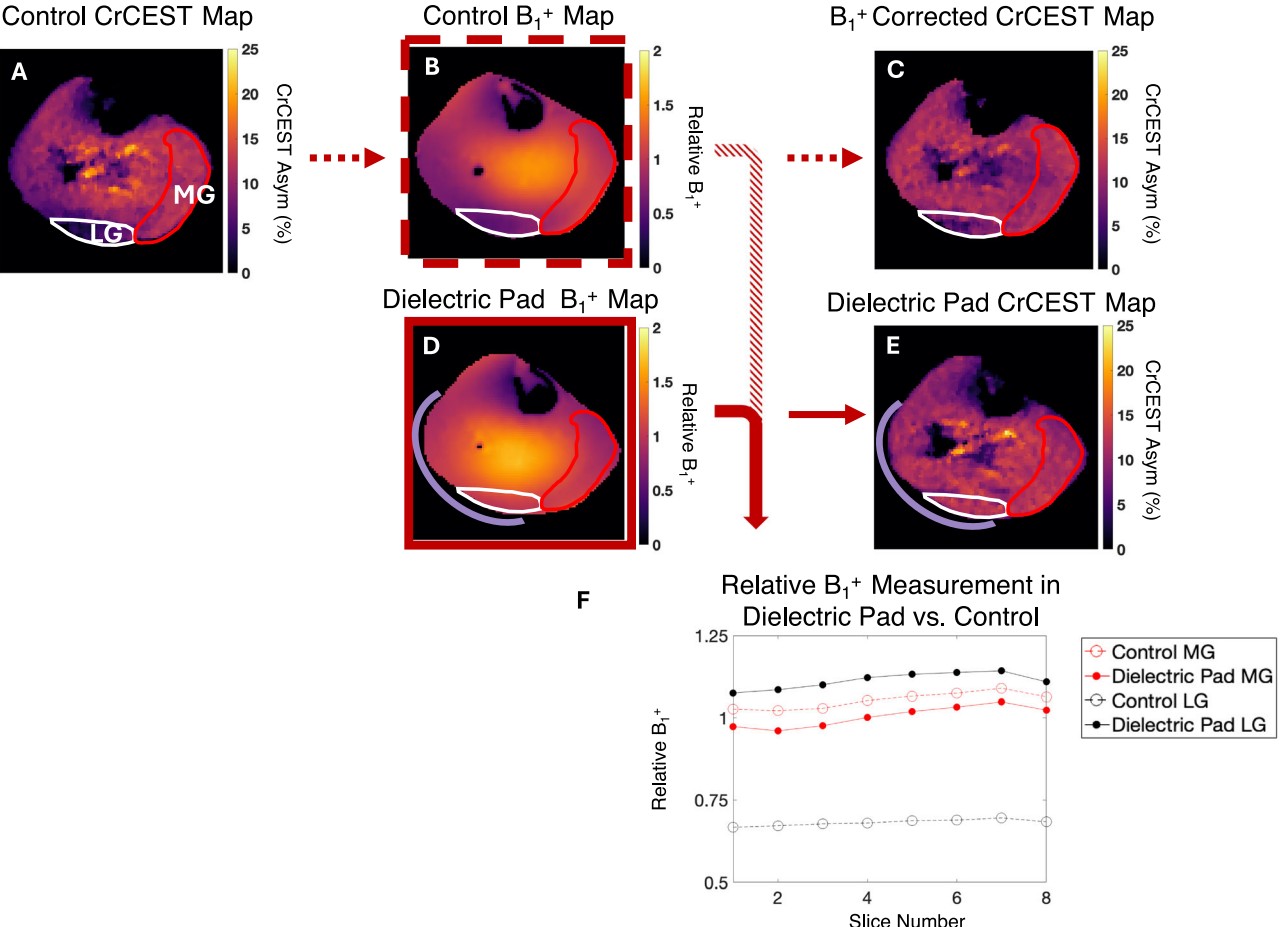

**Fig. 3 | B₁⁺ correction using linear correction vs. dielectric pads to improve CrCEST. A** The control CrCEST map without any $B_1^+$ correction is shown with ROIs drawn in the medial (MG, red) and lateral gastrocnemius (LG, white). **B** A control $B_1^+$ map with units of relative $B_1^+$ displays inhomogeneity in the LG. **C** Linear corrected $B_1^+$ CrCEST maps display moderately recovered signal in LG. **D** $B_1^+$ map acquired after placement of a dielectric pad ($18 \times 18$ cm²) under the lateral gastrocnemius which is represented by the purple crescent shape. The dielectric pad enhances the relative $B_1^+$ and improves $B_1^+$ homogeneity. **E** CrCEST map acquired with a dielectric pad and without $B_1^+$ correction. The CrCEST signal is recovered in the LG. **F** The relative $B_1^+$ values of each ROI in **B** and **D** are plotted. **B**, **D** represent the third slice, but values for each slice are reported. Open data-points correspond to measurements taken without dielectric pads and solid data-points correspond to measurements with dielectric pads. Source data are provided in Source Data file.

association between recovery time and change in pH was not statistically significant ($\rho = 0.13$, $p = 0.45$). There is not evidence of a significant relationship between recovery time and change in pH for the mild exercise, likely because, by design, the change in pH was so minimal. Figure 6B shows that during moderate/intense exercise between $\tau_{Cr}$ becomes strongly and positively correlated to pH ($\rho = 0.67$).

For the moderate/intense exercise group, the correlation between changes in recovery time and change in pH was statistically significant ($p$ value = 0.001). There is sufficient evidence to conclude a significant relationship between $\tau_{Cr}$-values and change in pH for the moderate/intense exercise group. Furthermore, there was a statistically significant difference in recovery time ($p < 0.001$) and change in pH ($p < 0.001$) between the two exercise groups. Participants in the moderate/intense exercise group tended to have higher change in pH and longer recovery rate.

## Discussion

Our research findings indicate that the occurrence of intracellular acidosis results in delayed recovery of $\tau_{Cr}$. Consequently, the $\tau_{Cr}$-values obtained in the presence of intracellular acidosis are distorted and may not accurately represent the true OXPHOS capacity. As deconvolving the pH effect would require participant-specific quantitation of pH

recovery and complicated mathematical modeling, it is more practical to prescribe a mild exercise stimulus that does not lead to significant pH changes. With the prescribed mild exercise stimulus, when a pH change of ≤0.1 units was achieved, the post-exercise CrCEST elevation was sufficiently high (>10%), and the recovery time constant sufficiently long to be detected by our CrCEST MRI protocol.

Conventionally, the resistance for plantar flexion exercise has been established according to each individual's MVC. In this study, we empirically set criteria for mild exercise set in terms of resistance level (% of MVC), pedal-pushing frequency, and duration as a starting place for designating exercise stimulus intensity. The criteria were subsequently fine-tuned on an individual basis by taking into account both the post-exercise pH shifts and CrCEST asymmetry value. Notably, 22% of participants exhibited a post-exercise pH drop exceeding 0.1 when subjected to a resistance level of 10% MVC and a pedal-pushing frequency of 20. Additionally, one participant had a CrCEST Asym. increase of <10% after mild exercise. This shows that MVC, by itself, is not robust enough to ensure all participants have pH shifts ≤0.1 and sufficient CrCEST enhancement.

Amongst the majority of participants, mild exercise $\tau_{Cr}$-values were <100 s with the exception of three participants having values >100 s. Surprisingly, even moderate/intense exercise resulted in eight participants with $\tau_{Cr}$-values < 100 s despite having pH shifts >0.1 units.

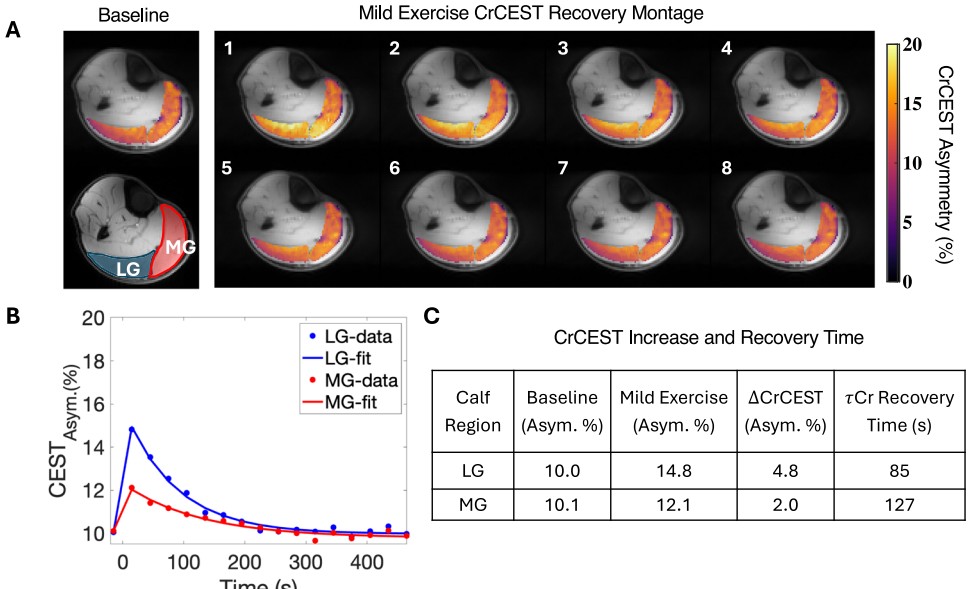

**Fig. 4 | Mild Exercise CrCEST Recovery and the $\tau_{Cr}$ recovery time(s). A** The baseline CrCEST map is displayed in the first column. Beneath the CrCEST map are segmentations of the lateral (LG) and medial gastrocnemius (MG) which are represented by the red and blue ROIs. In the next column, the image labeled 1 is the first CrCEST map post exercise. The time course of the CrCEST recovery flows from left to right and then continues in image 5 from left to right. Only the first eight CrCEST maps are displayed. All slices correspond to the sixth slice, out of the eight acquired. **B** The post-exercise CrCEST elevation and the subsequent creatine recovery that occurred in both ROIs are graphed in blue and red datapoints which correspond to the LG and MG, respectively. The solid line is the fit of the exponential decay back to baseline for the LG (blue) and MG (red). **C** A table of the baseline, post-exercise CrCEST, and ΔCrCEST asymmetry values, as well as the $\tau_{Cr}$ recovery time (s) is shown. Source data are provided in Source Data file.

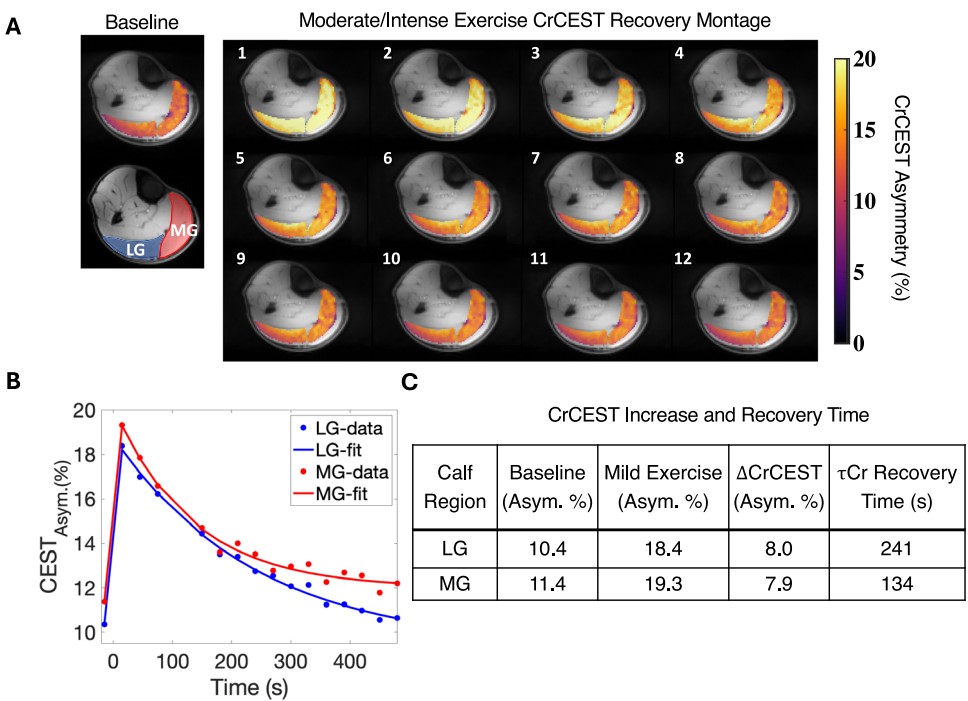

**Fig. 5 | Moderate/intense exercise CrCEST recovery and the $\tau_{Cr}$ recovery time(s). A** The first panel displays the baseline and segmented portions of the lateral (LG) and medial gastrocnemius (MG) in blue and red, respectively. The second panel contains CrCEST recovery maps. The first image was acquired 30 s post-exercise, and all subsequent images are 30 seconds apart. **B** The recovery of the CrCEST signal back to baseline are plotted. Each datapoint is a mean of the CrCEST signal within the respective ROI where blue and red datapoints reflect the measured CrCEST values in the LG and MG regions, respectively. The blue and red lines are the exponential fits to the measured data for the LG and MG regions, respectively. **C** The CrCEST asymmetry values are shown for baseline, post-intense exercise, and ΔCrCEST. The last column of **C** shows the creatine recovery time. Source data are provided in Source Data file.

**Table 1 | CrCEST Asym. % measurements during mild and moderate/intense exercise**

| Participants | Exercise regime | Calf region | CrCEST (Asym %) | | | |
| --- | --- | --- | --- | --- | --- | --- |
| | | | Baseline | Post exercise | ΔCrCEST | $\tau_{Cr}$ Recovery time (s) |
| 25 | Mild | LG | 10.3 (±1.8) | 15.0 (±3.8) | 4.7 (±2.7) | 62 (18–229) |
| 18 | Mild | MG | 10.9 (±1.3) | 14.6 (±2.8) | 3.7 (±2.1) | 53 (16–168) |
| 17 | Moderate/intense | LG | 9.9 (±1.9) | 16.9 (±3.7) | 7.0 (±2.6) | 128 (53–750) |
| 20 | Moderate/intense | MG | 10.9 (±1.2) | 17.6 (±2.4) | 6.7 (±1.6) | 109 (39–548) |

Columns two and three specify the exercise stimulus and ROI region for each measurement. The baseline, post-exercise, and ΔCrCEST Asym. %, as well as the $\tau_{Cr}$, are shown in the last four columns. For each moderate/intense exercise protocol, CrCEST was measured in both calf regions whereas only one carnosine acquisition was taken in the most acidic portion of the calf as determined by the mild exercise. Source data are provided in Source Data file.

**Table 2 | pH measurements during mild and moderate/intense exercise**

| Participants | Exercise regime | Calf region | C2-H Peak Position (ppm) | | pH | | |
| --- | --- | --- | --- | --- | --- | --- | --- |
| | | | Baseline | Post exercise | Baseline | Post exercise | ΔpH |
| 22 | Mild | LG | 8.04 (±0.04) | 8.04 (±0.03) | 6.95 (±0.07) | 6.93 (±0.06) | 0.03 (±0.04) |
| 19 | Mild | MG | 8.02 (±0.02) | 8.03 (±0.03) | 6.98 (±0.05) | 6.97 (±0.07) | 0.03 (±0.03) |
| 10 | Moderate/intense | LG | 8.02 (±0.04) | 8.19 (±0.11) | 6.99 (±0.09) | 6.64 (±0.22) | 0.36 (±0.21) |
| 10 | Moderate/intense | MG | 8.02 (±0.02) | 8.17 (±0.10) | 6.98 (±0.04) | 6.69 (±0.20) | 0.30 (±0.21) |

The means and standard deviations of the baseline ppm, post-exercise ppm, baseline pH, post-exercise pH, and ΔpH are shown in columns four through eight. The criteria enforced for a mild exercise required a pH shift ≤0.1 units. Column two shows which exercise stimulus was being measured and column three specifies the voxel location. Source data are provided in Source Data file.

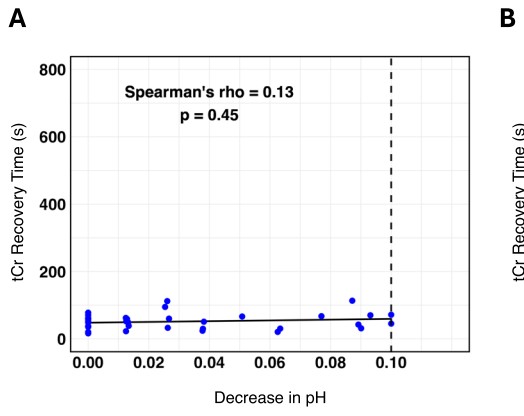
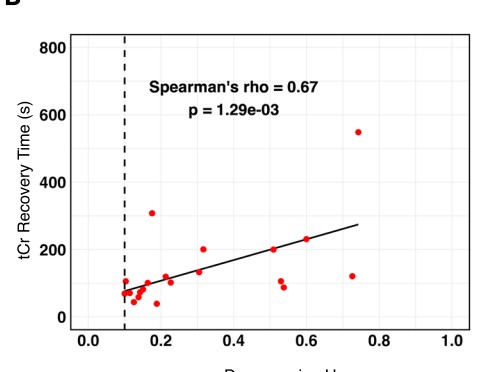

**Fig. 6 | Plot of decrease in pH vs $\tau_{Cr}$ for mild and moderate/intense exercise bouts. A, B** The decrease in pH vs $\tau_{Cr}$ is measured with respect to mild (blue datapoints, n = 37) (**A**), and moderate/intense (**B**) exercise (red datapoints, n = 20). **A** displays a black dotted line at y = 0.1 which was the threshold for the mild exercise regime. In each figure the line of best fit is shown. On each graph is the result of the Spearman's coefficient (ρ) and p value (two-sided t test) of each correlation. Source data are provided in Source Data file.

However, seven of these eight participants had shifts <0.15 which suggests their exercise may have been more moderate exercise, with less acidosis, which led to more rapid $\tau_{Cr}$-values. These seven measurements were in this intermediate region, 0.1–0.15 pH unit shift, and thus are towards the more moderate end of the moderate/intense exercise stimulus spectrum.

[31]P-MRS has conventionally been the most popular noninvasive method for evaluating OXPHOS capacity of exercised muscle, in addition to the advantage of being able to obtain simultaneous pH assessments; thus, it has proven successful in making muscle group specific measurements of $\tau_{PCr}$-values. In this study, [1]H-MRS was utilized for deducing intracellular pH changes by observing shifts in carnosine's C2-H peaks associated with pH variations. C2 and C4 protons of carnosine have physiological pKa's and are sensitive to changes in pH[18,19]. Their sensitivity to pH correlates well with [31]P results and lactate measurements[20–22]. [1]H-MRS was preferred for post-exercise intracellular pH measurement over [31]P spectroscopy for multiple reasons: (1) due to the abundance of [1]H in the body and its larger gyromagnetic

ratio, the sensitivity of CrCEST is three orders of magnitude greater[14], (2) spatial localization utilizing volume coils allows for muscle-specific analysis whereas conventional [31]P-MRS employs surface coils which receive signal from all muscle regions, (3) the effects of $B_1^+$ are compensated for better because we utilize a volume coil which enables a more homogenous $B_1^+$ which is further improved by use of a dielectric pad, (4) there is seamless transition between pH and CrCEST measurement without the need for any coil changes or experimental setup modifications, and (5) retrospective motion correction can be applied to this technique to correct for participant movement during the post-exercise acquisition. Due to the major spatial resolution issues and RF coil sensitivity differences, it is difficult to compare the recovery rates or OXPHOS measures obtained from [31]P-MRS with those from CrCEST MRI.

We examined the relationship between recovery time and the change in pH shift. The Spearman correlation coefficient (ρ) was calculated to assess the strength and direction of this association. $\tau_{Cr}$ and the change in pH shift were weakly and positively correlated. Results

from the mild exercise regime show that pH does not significantly influence the $\tau_{Cr}$, since the tissue is not burdened by lactic acid produced via glycolytic pathways. This likely means that the CrCEST signal more consistently represents OXPHOS measurement without confounding pH issues. In contrast, with the moderate/intense exercise stimulus (Fig. 5B) $\tau_{Cr}$ is strongly and positively correlated with pH. As pH increases, the CrCEST signal begins to increase monotonically showing that the measurement may become less accurate in reflecting OXPHOS as glycolytic activity increases. Therefore, these exercise techniques can be employed to study oxidative metabolism in mild exercise conditions and glycolytic metabolism in moderate/intense exercise.

The mean pH shift was similar in both the LG and MG for mild exercise, yet the $\tau_{Cr}$-values showed that the LG was preferentially utilized in exercise compared to the MG. This is additional evidence to show that during mild exercise, CrCEST-derived $\tau_{Cr}$ and pH are not inextricably linked.

Since the primary aim of this study was to accurately measure $\tau_{Cr}$-values, it is crucial to detect the temporal trend accurately. $B_1^+$ inhomogeneities lead to variation in saturation power across muscle groups with a drastic drop in gastrocnemius region, where relative $B_1$ could be -0.4–0.65. In our earlier study[15], to account for $B_1^+$ variations, we collected multiple saturation power datasets to account for $B_1^+$ field variations. However, to make this application more clinically appropriate, it is important to decrease scan time. To decrease scan time and account for $B_1^+$ variations, we positioned a dielectric pad beneath the lateral gastrocnemius muscle to enhance the relative $B_1^+$ field (Fig. 3).

Limitations to this method are that the mild exercise measurements of carnosine and CrCEST need to be acquired across two bouts, which may be susceptible to variations across each bout. Methods to reduce number of exercise bouts will further improve the precision of the measurement.

Additionally, six participants worked too hard and had pH shifts >0.1. To combat this limitation, we have recently implemented a program that reads the spectra from the scanner and calculates the pH shift in each muscle group in real time. This occurs during the -10 minute recovery period and if the mild exercise leads to pH > 0.1 drop then the scanner can decrease the psi experienced by the ergometer before acquiring CrCEST.

An unavoidable limitation to this method is participant movement. Though our method is resilient to minor motions post-exercise, large translations of the calf lead to uncorrectable spectra and CrCEST data. Therefore, motion detection algorithms should be employed to preserve data fidelity.

Carnosine $^1$H-MRS can be used to quickly measure pH change induced by a prescribed exercise regime and thus can facilitate an unbiased measurement of personalized muscle group specific OXPHOS capacity with CrCEST. It also enables the measurement of muscle pH prior to exercise and thus determines the potential muscle acidosis conditions. Overall, this comprehensive metabolic imaging protocol will enable personalized, muscle-specific OXPHOS capacity measurements in both healthy aging and myriad other disease states impacting muscle mitochondria.

## Methods
### Study design
Twenty-seven participants (14 males, 13 females; mean age 27 (±5) years, range 23–44) were enrolled in a protocol approved by the Institutional Review Board of the University of Pennsylvania, Philadelphia, PA 19104, and written informed consent was obtained and participants were compensated for participation. Participant's sex was determined based on self-reporting. Although our enrollment was balanced with regards to sex and gender, specific analysis on sex and gender was not performed given the small sample size of each exercise group. Prior to scanning, each participant's MVC was measured as

described below. Ten healthy volunteers were scanned using only the mild exercise stimulus (7 males, 3 females; mean age 30 ± 7, range 23-44) and 17 healthy volunteers (7 males, 10 females; mean age 27 ± 3, range 23–34) were scanned using both mild and moderate/intense stimuli. All MR images and spectra were acquired at a 7 T MRI scanner (MAGNETOM Terra, Siemens Healthcare, Erlangen, Germany) using a 28-Channel phased-array knee coil (Quality Electrodynamics, Mayfield Village, USA). Two different types of scanning sessions were performed. The first session tested the robustness of achieving an optimal mild exercise ($n = 10$). This mild exercise was optimized by setting an upper and lower bound which is described further in the section below, 'Defining exercise stimuli'. In the second session, participants underwent the mild exercise stimulus for the first bout, and then had a second moderate/intense exercise ($n = 17$). Both the mild stimulus and moderate/intense exercise stimulus were used to test the dependence of $\tau_{Cr}$ on pH shift.

A tri-plane localizer and anatomical multi-slice axial images were first acquired to find the ROI, which included the medial and lateral gastrocnemius. Manual shimming was performed for spectroscopy voxels to achieve values <40 Hz for all participants. The shim volume for spectroscopy and CrCEST were the same, the entire calf was covered in medial-lateral and anterior-posterior directions and the superior-inferior direction was always 40 mm (i.e., 8 slices × 5 mm each). Baseline carnosine spectra data were acquired for 2 minutes, followed by performing mild exercise bouts and assessing pH shift using carnosine spectroscopy. A 10-minute rest period was allotted to allow pH recovery, as determined empirically (Supplemental Fig. 2). Subsequently, the baseline CrCEST protocol was run for 5 minutes which included a $B_0$ map[24], $B_1^+$ map[25], and creatine-weighted CEST acquisition. This allowed a total rest time of 15 minutes before the second mild exercise bout. Following the second exercise, post-exercise $B_0$ map, $B_1^+$ map and CrCEST acquisitions lasted for 11 minutes. The protocol is depicted as a flowchart in Supplemental Fig. 3.

### Measuring MVC
Participants were positioned supine on a medical bed and pushed against a loadsol (novel electronics inc., St. Paul, MN, USA) fastened to a wall with maximal force for 4–5 s using only the calf muscle. A harness fastened to the bed was worn by each participant to avoid body movement during contraction. The setup is shown in Supplemental Fig. 4. Once firmly fastened, the loadsol was zeroed and the participant was coached to isolate the contraction to their gastrocnemius and not to lift the hips. For each participant's MVC measurement, five measurements were taken, the minimum and maximum values were excluded and the average of the three remaining attempts was taken to calculate the MVC. To ensure maximal contraction, participants were encouraged with each attempt. Each measurement was recorded in Newtons and then converted into pounds per square inch (PSI).

### Defining exercise stimuli and optimizing participant in-magnet setup
An MR compatible pneumatic ergometer (Trispect; Ergospect, Innsbruck, Austria) was used for in-magnet exercise. The device has two ports, one for pressurized air and another to attach a vacuum line. The vacuum line is helpful in assuring ergometer and gantry stability. In addition, the setup employed a deuterium-doped calcium titanate ($CaTiO_3$) dielectric pad ($18 \times 18$ cm$^2$), which was placed under the lateral gastrocnemius to improve $B_1^+$ field homogeneity (7TNS, Multiwave Imaging, Marseille, France).

The mild and moderate/intense plantar flexion exercise stimulus included specifications for the following: resistance level (in psi), push pedal frequency, and exercise duration. To determine the optimal exercise stimulus intensities, we had four participants perform a two-minute plantar flexion exercise within the MRI scanner under varying combinations of resistance levels (i.e., % of MVC) and pedal-pushing

frequencies. As a preliminary criterion for designating mild exercise, we aimed to identify an exercise stimulus that could be accomplished without any associated discomfort or muscle burning sensation associated with lactate formation due to pH drop. After experimentation, we designated the mild exercise stimulus as follows: resistance set to 10% MVC, push pedal frequency to 20 beats-per-minute (BPM), a plantar flexion angle of 60 degrees, and a total exercise duration of 2 minutes. To optimize the mild exercise an upper and lower bound was established to ensure an exercise that does not generate too much lactate, yet also yields sufficient free creatine. The upper bound was determined by assessing pH shift using carnosine spectroscopy, if the pH shift was ≤0.1 then the exercise was too demanding. Whereas the lower bound depended on the mild exercise to induce an increase in the CrCEST Asym. % measurement by at least 10% of the baseline value. Moderate/intense exercise stimulus was 20% MVC, push pedal frequency to 30 BPM, and exercise until exhaustion. This included a lower bound which depended on a pH change of >0.1 units.

Additionally, in the mild and moderate/intense exercise scanning session, to determine which muscle group was utilized most during exercise, the pH shift was calculated for each gastrocnemius muscle group during the mild exercise bout. The muscle group with the largest pH shift was also measured during the moderate/intense exercise bout. The MRI scanning for the moderate/intense exercise bout consisted of baseline CrCEST, moderate/intense exercise, and then post-exercise CrCEST and carnosine acquisition. The post-exercise acquisition consisted of three CrCEST acquisitions (~1 m 40 s), a carnosine acquisition in the muscle group that had been previously identified to be most exercised (22 s), and then 12 more CrCEST acquisitions (6 m 10 s). pH measurements were delayed ~1 m 40 s for two reasons: (1) healthy participants show $\tau_{Cr}$ of ~2 minute test which means CrCEST acquisition immediately after exercise is essential for accurate fitting, (2) empirically, we found that pH recovery does not start immediately post-exercise. Instead, the pH remains constant or continues to become more acidic for ~1–2 mins post exercise which has also been demonstrated in other work[23,26]. A post-exercise $B_0$ map (2 m), $B_1^+$ map (25 s), and reference image (15 s) were also acquired.

### Technical parameters for imaging and spectroscopy sequences

The CrCEST sequence consisted of a pulse train (5 × 99 ms Hanning windowed, duty cycle 99%, $B_{1,rms}$ of 2.9 μT), followed by a centric-ordered GRE readout with FOV = 160 × 160 × 40 mm³, resolution = 1.4 × 1.4 × 5 mm³, TR = 3.5 ms, TE = 1.47 ms, receiver BW = 710 Hz/pixel. Using a power of $B_{1,rms}$ of 2.9 μT has been shown to provide the best saturation efficiency as it is the power at which both CrCEST Asym % signal and SNR of the anatomical image can be simultaneously maximized[14].

To ensure that the measured creatine signal comes primarily via chemical exchange and not cross relaxation pathways, a transient nuclear Overhauser effect (tNOE) experiment was performed on a phantom containing two separate vials. One contained deionized water and the other contained 100 mM creatine in DPBS, each had a pH of 7.05 and were scanned at room temperature. The same parameters were used as in previous work[27].

Other imaging parameters and the procedure for water saturation shift referencing imaging (WASSR) and $B_0$-map correction has been described previously[15]. The reference voltage was kept at 300 V for $B_0$-map, $B_1^+$-map and CEST-maps for each participant, as that voltage has been used in previous work with sufficient relative $B_1^+$, between 0.8 and 1.1, in the gastrocnemius[15]. To suppress the fat signal, a chemical shift-selective fat saturation pulse was applied immediately prior to image readout[15]. Two dummy shots were performed for CEST, WASSR, and $B_1^+$ map sequences to ensure $M_z$-value stayed the same before every shot[15]. To ensure starting acquisition immediately after exercise, the dummy shots and exercise were synchronized to complete simultaneously. Furthermore, the fourth timepoint post moderate/intense

exercise also included two dummy shots which contributed to the 45 s gap between the third and fourth timepoint.

WASSR-weighted images, corresponding to frequency offsets from ± 0 to ± 0.9 ppm with a step-size of ± 0.15 ppm and scan time of 80 s, were acquired with a saturation pulse train (2 × 99 ms pulse duration, 99% duty cycle, $B_{1rms}$ of 0.29 μT). Data for generating a $B_1^+$ map were obtained using a flip crush pulse sequence[25] with three flip angles (20°, 40°, and 80°). All other imaging parameters were identical to CEST imaging and it took approximately 25 s to acquire a $B_1^+$ imaging data set which included two dummy shots preceding the sequence. Data corresponding to the 40° and 80° flip angles are sensitive to detect lower $B_1^+$ values (relative $B_1^+$ < 1.0); whereas 20° and 40° flip angles are sensitive to higher $B_1^+$ (relative $B_1^+$ > 1.0)[15].

A spectrally selective 90° E-BURP pulse centered at 7.7 ppm with an excitation bandwidth of 2 ppm was used to excite the carnosine peaks (TR/TE: 1400/18 ms, 16 averages), and three orthogonal, narrow-band spatially selective refocusing 180° Shinnar-Le Roux (SLR) pulses (BW: 800 Hz) were used for localization[28]. We used spectrally selective excitation as opposed to more conventional broadband excitation because we found the spectra had more resolvable carnosine peaks. VAPOR[29] water suppression was employed with a bandwidth of 150 Hz. This suppresses not only water but also any cross-relaxing peaks (i.e., NAD⁺, ATP) which occur in the downfield spectrum. Other parameters included receiver bandwidth 2000 Hz, 16-step phase cycling, voxel size 30 × 15 × 15 mm³, and temporal resolution ~22 s. Baseline scans involved three averages in both the MG and LG. Post-mild exercise scans included one transient in each ROI, while the post moderate/intense exercise scan involved one transient in one of the muscle group ROIs. The ROI selected for the post moderate/intense exercise scan was the muscle compartment with the highest level of acidosis observed during the mild exercise. An unsuppressed water scan in each ROI was also acquired pre- and post-exercise; however, these were not used for any analysis. A reference voltage calibration was implemented for each ROI prior to ¹H-MRS acquisitions as performed in previous work[30].

### CrCEST image analysis and carnosine spectroscopy postprocessing

Data processing was performed using in-house MATLAB scripts. Acquired CEST-weighted images were corrected for $B_0$ inhomogeneities. To perform the interpolation needed for $B_0$-correction, we fit the raw CEST-weighted images to a second-order polynomial. Then, the CEST asymmetry ratio was computed from CEST-weighted images at frequency offset of ($\Delta\omega$ = +1.8 ppm) using the following equation:

$$CEST_{asym} = \frac{M_{Sat}(-\triangle\omega) - M_{Sat}(\triangle\omega)}{M_{Sat}(-\triangle\omega)} \quad (2)$$

where $M_{Sat}$ (±Δω) is the image signal following a train of saturation pulses applied at frequency offsets ±Δω relative to water peak. For all participants, the slice with the highest CrCEST Asym. % was selected.

The $B_0$ map generated using WASSR imaging was used to perform $B_0$-correction of CEST-weighted images[15]. Although $B_1^+$ maps were obtained, $B_1^+$ correction was not performed to account for $B_1^+$-inhomogeneity. Instead, placement of a dielectric pad on the lateral gastrocnemius helped to improve relative $B_1^+$ inhomogeneity Supplemental Fig. 5. It is important to note that the CrCEST maps in Figs. 1 and 3–5 are denoised using MATLAB's median filter in the image processing toolbox. This was used solely for graphical representation and the median filter was not employed on the CrCEST data before analysis. The $\tau_{Cr}$ recovery data for mild and moderate/intense exercise was fit using a mono-exponential decay.

With regard to ¹H-MRS, each spectrum was frequency corrected by setting the downfield lipid peak to be constant at 5.56ppm, the lipid signal was then removed using Hankel singular value decomposition,

and the spectrum was phase corrected using an in-house MATLAB script[31]. The pH shift was calculated via the following adjusted Henderson-Hasselbach equation[20,32]:

$$pH = pK_a + \log\left(\frac{H_A - \sigma_{C2H}}{\sigma_{C2H} - H_B}\right) \quad (3)$$

The C2 proton $pK_a$, maximum acidic shift ($H_A$), and maximum basic shift ($H_B$) are 6.81, 8.57, and 7.65, respectively[32,33]. The measured C2-H proton peak is represented as $\sigma_{C2H}$. The C2 proton was used for the adjusted Henderson-Hasselbach equation because the slope of its titration curve is less steep and thus more sensitive than the C4 proton at 7 ppm.

## Statistical analysis

Descriptive statistics including mean, median, and range were measured in the 27 participants. The Spearman correlation coefficient ($\rho$) was calculated to assess the strength and direction of the association between $\tau_{Cr}$-values and change in pH. The $p$ values for this two-sided test are also provided. Participants that had $\tau_{Cr}$-values and pH shifts that were greater than 2.5 standard deviations from the mean were considered outliers and removed from analysis. The Wilcoxon rank sum test was used to compare the $\tau_{Cr}$ and change in pH between the mild and moderate/intense exercise bouts.

## Data availability

The data that support the findings of this study are available from the corresponding author upon request. Requests for the data should be submitted to: Dr. Ravinder Reddy via krr@pennmedicine.upenn.edu. Please expect a two-week response time. Source data are provided with this paper.

## Code availability

The CrCEST MRI acquisition sequence used in the current study is developed at the Center for Advanced Metabolic Imaging in Precision Medicine at the University of Pennsylvania and is available for use at other institutions on compatible Siemens scanners by Consumer-to-Producer (C2P) agreement. Postprocessing code will be shared upon request. Requests for the code should be submitted to: Dr. Ravinder Reddy via krr@pennmedicine.upenn.edu. Please expect a two-week response time.

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

## Acknowledgements

The image of a contracted gastrocnemius was reused with permissions from Complete Anatomy (www.3d4medical.com), copyright 3D4Medical LTD, 2021. Research reported in this publication was supported by the National Institute of Biomedical Imaging and Bioengineering of the National Institutes of Health under award Number P41EB029460 (R.R.), R03EB030663 (D.K.), and T32EB020087 (Felix Wehrli, University of Pennsylvania, Philadelphia, PA 19104), as well as the National Institute of Aging under award number R56AG062665 (R.R.) and R01AG071725 (R.R.).

## Author contributions

Conceptualization: R.R. Methodology: R.R.A, D.K, N.W, and R.R. Investigation: S.M, A.C, R.R. Formal analysis: R.R.A, D.K., B.B., P.J., A.K., F.L., R.P.R.N., and N.W. Data curation: R.R.A., D.K, F.L., and N.W. Visualization: R.R.A., D.K., F.L., and N.W. Supervision—D.K., N.W., and R.R. Original draft—R.R.A, D.K, and R.R. review and editing- R.R.A, D.K., P.J, F.L., R.P.R.N., S.M, A.C, N.W., R.R.

## Competing interests

The authors declare no competing interests.
