## [Peer Review File · Nature Communications]

REVIEWER COMMENTS

Reviewer #1 (Remarks to the Author):

This manuscript reports a CrCEST 1H-MR imaging method for measuring OXPHOS capacity (in terms of TCr) instead of using conventional 31P-MRS (in the form of TPCr). They also used carnosine spectroscopy as a separate scan in a protocol for measuring and detecting shifts in muscle pH during exercise bouts.

Both of these methods were reported in earlier literature separately, but their tandem use is uniquely reported in this manuscript to measure OXPHOS capacity.

The manuscript is very well written with very few grammatical errors. Specifically:

1. Page 2, line 64, add Phosphocreatine
2. Comment on exchange rate (K_{sw}), saturation efficiency and proton transfer ratio of CrCEST method.
3. Comment on relative contributions of chemical exchange and cross relaxation pathways for CrCEST experiments.
4. For moderate/intense exercise CrCEST recovery, post exercise carnosine acquisition was acquired in the middle of CEST experiment (after 3 timepoints). Why? This is not clear. Why after 3rd timepoint and not 1st timepoint?
5. How is the CrCEST method different or efficient from the conventional 31P-MRS method for measuring OXPHOS capacity in terms of sensitivity, motion of participants and B1+ inhomogeneity?

Reviewer #2 (Remarks to the Author):

In order to better understand the relationship between the muscle-specific creatine (Cr) recovery time (τ_{Cr}) and the intracellular acidosis, authors have investigated 3D creatine chemical exchange by saturation transfer (CrCEST) in combination with 1H-MR spectroscopy after mild and moderate/intense exercise stimuli and determined the exercise stimulus as a proportion of the individual-specific maximal voluntary contraction (MVC) that achieved non-significant and significant changes in pH for mild and moderate/intense exercise, respectively. Finally, the dependence of CrCEST recovery time on pH using

the different exercise stimuli was determined. The manuscript is well written and the results are convincing. However, a revision is recommended due to few minor concerns as follows: 1) narrow band MR spectroscopy was used to detect the imidazole proton resonances at 7 and 8 ppm. While pH is going down to acidic level, inclusion of lactate as reported in earlier studies (reference#19) using non-selective excitation of 10ppm along the spectral dimension may be more convincing. Hence, authors should discuss why a narrow spectral region was excited. 2) one annotation in Fig.1 may be changed from 'anatomic image' to 'anatomical images'. 3) pp7: A reference should be added while referring to the baseline ppm and pH. 4) it is unclear what 'optimal mild exercise' means. 5) pp11: the last sentence of the 1st para should be revised also. 6) Pp12: the 2nd sentence in the 2nd para should be revised also. 7) it is unclear if two additional dummy shots were added to the base 4 dummy scans by default. 8) 'Proton MRS (1H-MRS)' or 1H-MRS should be used consistently.

Reviewer #2 (Remarks on code availability):

A major focus is on CrCEST and MR Spectroscopy to investigate pH and creatine recovery time (τ_{Cr}). So the code is partially reflective of the actual work completed in this manuscript.

Personalized and Muscle Specific OXPHOS Measurement with Integrated CrCEST MRI and Proton MR Spectroscopy

Ryan R Armbruster^{1*}, Dushyant Kumar^{1*}, Blake Benyard¹, Paul Jacobs¹, Aditi Khandavilli², Liu Fang³, Ravi Prakash Reddy Nanga¹, Shana McCormack⁴, Anne R. Cappola⁵, Neil Wilson¹, and Ravinder Reddy¹
¹Center for Advanced Metabolic Imaging in Precision Medicine, University of Pennsylvania, Philadelphia, PA, USA ²Department of Biology, Department of Nutrition and Science, Cornell University, Ithaca, NY, USA ³Department of Biostatistics, Epidemiology, and Informatics Perelman School of Medicine, University of Pennsylvania, Philadelphia, PA, USA ⁴Neuroendocrine Center, Division of Endocrinology and Diabetes, Children's Hospital of Philadelphia, Philadelphia, PA, USA ⁵ Division of Endocrinology, Diabetes, and Metabolism, University of Pennsylvania, Philadelphia, PA, USA
(*R.R. Armbruster and D. Kumar contributed equally to this work.)

Reviewer Comments & Responses

Comments to Author:

Reviewer 1:

This manuscript reports a CrCEST 1H-MR imaging method for measuring OXPHOS capacity (in terms of TCr) instead of using conventional 31P-MRS (in the form of TPCr). They also used carnosine spectroscopy as a separate scan in a protocol for measuring and detecting shifts in muscle pH during exercise bouts.

Both of these methods were reported in earlier literature separately, but their tandem use is uniquely reported in this manuscript to measure OXPHOS capacity.

The manuscript is very well written with very few grammatical errors. Specifically:

1. Page 2, line 64, add Phosphocreatine

Response: Correction made.

2. Comment on exchange rate (K_{sw}), saturation efficiency and proton transfer ratio of CrCEST method.

Response:

We thank the reviewer for these comments. These points were addressed in either the introduction (i.e. K_{sw}) or methods section (i.e. saturation efficiency). It has been shown that creatine's guanidinium protons exchange with bulk water at the rate of $950 \pm 100 \text{ s}^{-1}$ ¹².

Using a power of $B_{1,rms}$ of 2.9 μT has been shown to provide the best saturation efficiency as it is the power at which both CrCEST Asym % signal and SNR of the anatomical image can be optimally balanced ¹⁴.

3. Comment on relative contributions of chemical exchange and cross relaxation pathways for CrCEST experiments.

Response: In the manuscript we now address the contribution of cross relaxation to the CEST signal. This has been addressed in both the methods and results, as well as in the supplemental material sections.

To confirm that the measured creatine signal comes primarily via chemical exchange and not cross-relaxation pathways, a transient nuclear Overhauser effect (tNOE) experiment was performed on a phantom containing two separate vials. One contained deionized water and the other contained 100 mM creatine in DPBS, each had a pH of 7.05 and were scanned at room temperature. The same parameters were used as in previous work ²⁷.

Supplemental Fig. 1 shows that the cross relaxation contribution of the aliphatic protons of creatine, measured using transient nuclear Overhauser effect (tNOE), is negligible with a contrast of 0.07% from a 100 mM creatine phantom. Upon asymmetry analysis the contribution becomes 0.03%. Given that the physiological concentration of creatine is ~20-35 mM, this would mean that relative cross relaxation contribution to CrCEST signal would be <0.01%. This is consistent with CrCEST work which showed no appreciable CEST signal from the creatine methyl protons at -1.7 ppm compared to the guanadinium protons ¹².

4. For moderate/intense exercise CrCEST recovery, post exercise carnosine acquisition was acquired in the middle of CEST experiment (after 3 timepoints). Why? This is not clear. Why after 3rd timepoint and not 1st timepoint?

Response: We thank the reviewer for this point. We addressed this comment in our methods section.

pH measurements were delayed ~1m40s for two reasons: 1.) it is crucial to acquire CrCEST data immediately after exercise because initial datapoints are required to compute a robust τ_{Cr} values, 2.) empirically, we found that pH recovery does not start immediately post-exercise. Instead, the pH remains constant or continues to become more acidic for ~1-2 mins post exercise which has also been demonstrated in other work ^{23,26}.

Therefore, delaying the carnosine spectroscopy until after the third timepoint does not compromise the pH measurement, while enabling rapid CrCEST acquisitions for more accurate fitting.

5. How is the CrCEST method different or efficient from the conventional ³¹P-MRS method for measuring OXPHOS capacity in terms of sensitivity, motion of participants and B1+ inhomogeneity.

Response: These points addressed and elaborated on in the discussion section.

¹H-MRS was preferred over ³¹P-MRS for multiple reasons: 1.) due to the abundance of ¹H in the body and its larger gyromagnetic ratio, the sensitivity of CrCEST is three orders of magnitude greater than ³¹P-MRS ¹⁴, 2.) spatial localization utilizing volume coils allows for muscle specific analysis whereas conventional ³¹P-MRS employs surface coils which receive signal from all muscle regions within the penetration depth of the coil, 3.) there is a seamless transition between pH and CrCEST measurement without the need for any coil changes or experimental

setup modifications, and 4.) retrospective motion correction can be applied to this technique to correct for subject movement during the post-exercise acquisition.

Reviewer 2:

In order to better understand the relationship between the muscle-specific creatine (Cr) recovery time (τ_{Cr}) and the intracellular acidosis, authors have investigated 3D creatine chemical exchange by saturation transfer (CrCEST) in combination with 1H -MR spectroscopy after mild and moderate/intense exercise stimuli and determined the exercise stimulus as a proportion of the individual-specific maximal voluntary contraction (MVC) that achieved non-significant and significant changes in pH for mild and moderate/intense exercise, respectively. Finally, the dependence of CrCEST recovery time on pH using the different exercise stimuli was determined. The manuscript is well written and the results are convincing. However, a revision is recommended due to few minor concerns as follows:

1. Narrow band MR spectroscopy was used to detect the imidazole proton resonances at 7 and 8 ppm. While pH is going down to acidic level, inclusion of lactate as reported in earlier studies (reference#19) using non-selective excitation of 10ppm along the spectral dimension may be more convincing. Hence, authors should discuss why a narrow spectral region was excited.

Response: We thank the reviewer for their point.

Reference 19, now reference 20, uses carnosine for pH measurement similar to this paper. Reference 20, now reference 21, correlates lactate to pH measured from carnosine. Both papers use carnosine for pH measurement as this shows to be an accurate estimate of pH when compared to inorganic phosphate measured via ^{31}P -MRS. That is why we also used carnosine to measure pH. A spectrally selective excitation was employed because it resulted in the most resolvable carnosine peaks in <30s with minimal contribution from water and lipid signals. A conventional broadband sequence could have been used instead.

A sentence was added in the methods section.

2. One annotation in Fig.1 may be changed from 'anatomic image' to 'anatomical images'.

Response: Correction Made

3. pp7: A reference should be added while referring to the baseline ppm and pH.

Response: Correction Made.

4. it is unclear what 'optimal mild exercise' means.

Response: We thank the reviewer for this comment. We further clarified how optimal mild exercise is defined in the methods section titled 'Defining exercise stimuli and optimizing participant in-magnet setup'.

To optimize the mild exercise an upper and lower bound was established to ensure an exercise that does not generate too much lactate, yet also yields sufficient free creatine. The upper bound was determined by assessing pH shift using carnosine spectroscopy, if the pH shift was ≤ 0.1 then the exercise was too demanding. Whereas the lower bound depended on the mild exercise to induce an increase in the CrCEST Asym. % measurement by at least 10% of the baseline value.

5. pp11: the last sentence of the 1st para should be revised also.

Response: Correction made.

6. Pp12: the 2nd sentence in the 2nd para should be revised also.

Response: Correction made.

7. it is unclear if two additional dummy shots were added to the base 4 dummy scans by default.

Response: We thank the reviewer for this point. We clarified the language in our methods section.

Two dummy shots were performed for CEST, WASSR, and B_1^+ map sequences to ensure M_z -value stayed the same before every shot¹⁵. To ensure starting acquisition immediately after exercise, the dummy shots and exercise were synchronized to complete simultaneously. Furthermore, the fourth timepoint post moderate/intense exercise also included two dummy-shots which contributed to the 45s gap between the third and fourth timepoint.

8. 'Proton MRS (1H-MRS)' or 1H-MRS should be used consistently.

Response: Correction made.

(Remarks on code availability):

9. A major focus is on CrCEST and MR Spectroscopy to investigate pH and creatine recovery time (τ_{Cr}). So the code is partially reflective of the actual work completed in this manuscript. (Remarks on code availability): A major focus is on CrCEST and MR Spectroscopy to investigate pH and creatine recovery time (τ_{Cr}). So the code is partially reflective of the actual work completed in this manuscript.

Response: A Code availability section has been added to the end of the paper.

REVIEWERS' COMMENTS

Reviewer #1 (Remarks to the Author):

I thank the author's for responding to all of concerns and the manuscript is much improved. I have no additional suggestions.

Reviewer #2 (Remarks to the Author):

Using a 7T MRI scanner, authors have investigated 3D creatine chemical exchange by saturation transfer (CrCEST) in combination with ^1H MR Spectroscopy (MRS) after mild and moderate/intense exercise stimuli, and determined the exercise stimulus as a proportion of the individual specific maximal voluntary contraction (MVC) which achieved non-significant and significant changes in pH for mild and moderate/intense exercise, respectively. In response to earlier critiques, authors have revised the manuscript with a positive response.